# 7-Phenylheptanoic Acid-Hydroxypropyl β-Cyclodextrin Complex Slows the Progression of Renal Failure in Adenine-Induced Chronic Kidney Disease Mice

**DOI:** 10.3390/toxins16070316

**Published:** 2024-07-12

**Authors:** Kindness Lomotey Commey, Airi Enaka, Ryota Nakamura, Asami Yamamoto, Kenji Tsukigawa, Koji Nishi, Masaki Otagiri, Keishi Yamasaki

**Affiliations:** 1Faculty of Pharmaceutical Sciences, Sojo University, 4-22-1 Ikeda, Kumamoto 860-0082, Japan; kindnesslomotey@ph.sojo-u.ac.jp (K.L.C.); mmhat.enk831@gmail.com (A.E.); g1951079@m.sojo-u.ac.jp (R.N.); g2051122@m.sojo-u.ac.jp (A.Y.); tsukigawa@ph.sojo-u.ac.jp (K.T.); knishi@ph.sojo-u.ac.jp (K.N.); otagirim@ph.sojo-u.ac.jp (M.O.); 2DDS Research Institute, Sojo University, 4-22-1 Ikeda, Kumamoto 860-0082, Japan

**Keywords:** 7-phenylheptanoic acid, hydroxypropyl β-cyclodextrin, inclusion complex, CKD, adenine-induced CKD mice, tryptophan indole lyase, indoxyl sulfate

## Abstract

The characteristic accumulation of circulating uremic toxins, such as indoxyl sulfate (IS), in chronic kidney disease (CKD) further exacerbates the disease progression. The gut microbiota, particularly gut bacterial-specific enzymes, represents a selective and attractive target for suppressing uremic toxin production and slowing the progression of renal failure. This study investigates the role of 4-phenylbutyrate (PB) and structurally related compounds, which are speculated to possess renoprotective properties in suppressing IS production and slowing or reversing renal failure in CKD. In vitro enzyme kinetic studies showed that 7-phenylheptanoic acid (PH), a PB homologue, suppresses the tryptophan indole lyase (TIL)-catalyzed decomposition of tryptophan to indole, the precursor of IS. A hydroxypropyl β-cyclodextrin (HPβCD) inclusion complex formulation of PH was prepared to enhance its biopharmaceutical properties and to facilitate in vivo evaluation. Prophylactic oral administration of the PH-HPβCD complex formulation reduced circulating IS and attenuated the deterioration of renal function and tubulointerstitial fibrosis in adenine-induced CKD mice. Additionally, treatment of moderately advanced adenine-induced CKD mice with the formulation ameliorated renal failure, although tissue fibrosis was not improved. These findings suggest that PH-HPβCD can slow the progression of renal failure and may have implications for preventing or managing CKD, particularly in early-stage disease.

## 1. Introduction

Chronic kidney disease (CKD) refers to a gradual and progressive decline in kidney structure or function that persists for more than three months. CKD reduces the quality of life, increases the risk of premature mortality, and imposes a substantial economic burden on healthcare systems worldwide [1,2,3,4]. An essential feature of CKD is the accumulation of uremic toxins, including indoxyl sulfate (IS) and *p*-cresyl sulfate (PCS), in circulation, further exacerbating the disease progression [5,6,7]. Given its irreversible nature, current research efforts have focused on developing intervention strategies to delay the disease progression and improve associated health outcomes. One such approach has been to reduce the amount of these uremic toxins in circulation. This has been achieved using adsorbents such as AST-120 in patients with pre-dialysis and uremic-stage renal failure [8,9].

The gut microbiota has also emerged as a promising therapeutic target for managing CKD through the gut–kidney axis. This axis encompasses the bidirectional crosstalk between the gut microbiota and the kidneys, which is increasingly being recognized as relevant to the pathophysiology of CKD. Deteriorating renal function in CKD and its associated accumulation of circulating uremic toxins lead to gut microbiota dysbiosis. Meanwhile, gut microbial metabolism generates uremic toxins, thus establishing a vicious cycle [10,11,12]. Gut bacterial-specific enzymes, such as tryptophan indole lyase (TIL) (EC 4.1.99.1) and tyrosine transaminase (EC 2.6.1.5), are responsible for the initial stages of the biotransformation of the uremic toxin precursors indole and *p*-cresol. In the case of IS, the most extensively studied uremic toxin, tryptophan from dietary protein, is typically metabolized into indole through the action of TIL. Indole is then absorbed from the portal vein, converted into IS by sulfotransferases in the liver, and enters the circulation [13,14,15]. In CKD, IS builds up in circulation and the kidneys even in the early stages of the disease. Elevated IS has been shown to induce reactive oxygen species (ROS) and increase intravascular oxidative stress and renal tissue fibrosis [16,17,18]. We hypothesized that inhibiting TIL, which is responsible for the breakdown of dietary tryptophan, could reduce indole production and subsequently lower circulating levels of IS in CKD without inducing host toxicity. This is based on the fact that nearly all indole in the gut originates from the TIL-catalyzed degradation of dietary tryptophan and that TIL is not expressed in eukaryotes [13,19].

Short-chain fatty acids like propionate and butyrate, produced by bacterial fermentation of dietary fibers in the colon, are known to positively impact overall health, including safeguarding kidney health [20,21,22]. Interestingly, the phenyl derivative of butyrate, 4-phenylbutyrate (PB), has been shown to inhibit acute kidney injury in animal models [23]. In a recent study, PB and other low-molecular-weight chemical chaperones were shown to protect human renal proximal tubule epithelial cells by inhibiting endoplasmic reticulum (ER) stress. The study reported that the efficacy and potency of these compounds are attributable to the presence of a hydrophilic end followed by a long hydrocarbon, with the length of the hydrophobic hydrocarbon region correlating with potency [24]. These reports suggest that PB and its structurally related compounds may have the potential to preserve renal function in CKD. However, their effect on uremic toxins, particularly IS production and renal function in CKD, is unknown.

In the present study, we investigated the effect of PB and structural homologues 5-phenylvaleric acid (PV), 6-phenylcaproic acid (PC), and 7-phenylheptanoic acid (PH) on the activity of TIL in vitro. Based on the findings of the in vitro study, PH was selected for further in vivo evaluation. In the in vivo study, we assessed the potential of PH in reducing circulating IS levels and slowing or reversing the progression of renal failure in adenine-induced and 5/6 nephrectomized CKD mice [25,26]. PH, a viscous oil with limited aqueous solubility and an unpleasant odor, presented challenges for handling and oral administration. A hydroxypropyl β-cyclodextrin (HPβCD)-based solid system for PH was therefore prepared, allowing for the evaluation of its renal protective effects in vivo [27,28]. Our findings suggest that the PH-HPβCD complex formulation can slow renal failure in adenine-induced CKD mice partly by inhibiting gut TIL and reducing circulating IS, and may be useful for preventing or ameliorating renal failure during the early stages of CKD.

## 2. Results

### 2.1. PB and Structurally Related Compounds Inhibit Tryptophan Indole Lyase In Vitro

The effect of PB and structurally related compounds, PV, PC, and PH, on the conversion of tryptophan to indole was investigated by monitoring the activity of TIL in the presence of these compounds in vitro, as shown in Figure 1a–g. The decomposition of tryptophan to indole was indirectly monitored as the decrease in optical density in the reaction medium at 340 nm, as shown in Figure 1b, and was used to calculate the TIL reaction velocity. TIL reaction velocity was significantly reduced in the presence of the compounds (*p* < 0.05), with the length of the alkyl chain correlating positively with TIL inhibitory effects. In particular, PH, the longest chain homologue studied, produced a dose-dependent inhibitory effect on TIL activity (Figure 1c,d). Additionally, the depletion of tryptophan and accumulation of indole due to the action of TIL were directly monitored in the presence of PH. The results indicate that PH slowed the decomposition of tryptophan to indole (Figure 1e). Interestingly, *n*-butyric acid (BA) and *n*-heptanoic acid (HA), which are analogous to PB and PH, respectively, showed no TIL inhibitory effects (Figure 1f). This suggests that the aromatic ring is necessary for the observed inhibitory effects. Based on the results, PH was selected for further in vivo studies; however, owing to its less desirable biopharmaceutical properties, an HPβCD complex was prepared to facilitate the in vivo study. Therefore, the effect of HPβCD on the TIL-inhibitory activity of PH was also investigated in vitro. As shown in Figure 1g, the TIL-inhibitory activity of PH decreased in the presence of HPβCD (1:1) (*p* < 0.01). This decrease is likely due to competition between HPβCD and TIL for PH. Despite this competition, a significant TIL-inhibitory activity was still retained (*p* < 0.01). HPβCD alone showed no TIL-inhibitory activity.

### 2.2. Preparation and Characterization of PH-HPβCD Solid System

The biopharmaceutical limitations of PH (a poorly soluble viscous oil with an unpleasant odor) were addressed through CD complexation to produce an odorless solid preparation of PH with improved aqueous solubility. This enabled the evaluation of the potential renal protective effects of PH in vivo. The preparation and characterization of the PH-HPβCD solid system are illustrated in Figure 2a–d. A phase solubility study was conducted to select a suitable CD for this application. The study revealed that HPβCD demonstrated a higher complexation efficiency (CE) of 0.864 ± 0.027 and significantly enhanced the aqueous solubility of PH compared to natural βCD, which showed a CE of 0.604 ± 0.002 and provided only limited solubility improvement, as shown in Figure 2a. PH was lyophilized with HPβCD to obtain a free-flowing powder (PH content = 73.1 ± 1.0% w/w). The formulated PH appeared as fluffy sheets in the scanning electron micrograph (SEM) shown in Figure 2b. The solid system was analyzed using differential scanning calorimetry (DSC). A physical mixture of PH and HPβCD exhibited a curve with distinct sharp endothermic peaks at 149, 155, and 167 °C, as shown in Figure 2c. However, these peaks were absent from the DSC curve for the formulated PH-HPβCD solid system. Furthermore, the broad endothermal event, with a peak within the 90–130 °C range in the HPβCD curve, shifted to a lower temperature range and decreased intensity in the PH-HPβCD solid system. These suggest the possible formation of inclusion complexes in the prepared PH-HPβCD solid system. A dissolution study was also conducted on the prepared formulation. Figure 2d shows the dissolution profiles of PH alone and the prepared PH-HPβCD solid system. To analyze the dissolution profiles, the percentage of PH dissolved at 10 min (DP_10min_) and the dissolution efficiency at 60 min (DE_60min_) were calculated. The DP_10min_ values for PH alone and the prepared PH-HPβCD solid system were 17.15 ± 2.95% and 101.20 ± 2.12%, respectively (*p* < 0.001). The corresponding DE_60min_ values were 44.32 ± 2.89% and 96.35 ± 3.01%, respectively (*p* < 0.001).

### 2.3. Prophylactic Supplementation with PH-HPβCD Slowed the Progression of Renal Failure and Fibrosis in Adenine-Induced CKD Mice

To evaluate the effect of prophylactic supplementation with PH-HPβCD on the progression of CKD, adenine was used to induce renal failure in male C57BL/6J mice after acclimatization on a normal diet (standard chow for breeding) for a week (protocol shown in Figure 3a). A renal failure (RF) group was fed a diet containing 0.2% w/w adenine for three weeks. Two groups (RF + PH 5 and RF + PH 10) were fed diets containing 0.2% w/w adenine supplemented with the formulated PH-HPβCD solid system, equivalent to 5 mg and 10 mg/kg/day of PH, respectively. A control (Ctrl) group was fed a normal diet. Figure 3a–g shows the effects of the prophylactic supplementation with PH-HPβCD on the renal function of adenine-induced CKD mice. The RF group showed significant weight loss (*p* < 0.01) compared to the control group, as shown in Figure 3b. The RF mice also showed significantly lower food intake (*p* < 0.01) and higher water intake (*p* < 0.001), as shown in Appendix A. However, supplementation with PH-HPβCD showed a dose-dependent improvement in food intake (*p* < 0.05) and a strong tendency to attenuate the adenine-induced weight loss (Appendix A and Figure 3b). Plasma IS, blood urea nitrogen (BUN), and plasma creatinine (CRE) were significantly increased in the RF mice compared to the control group (*p* < 0.05), as shown in Figure 3c–e. On the other hand, the RF + PH 5 and RF + PH 10 mice showed significantly lower plasma IS (*p* < 0.05) and BUN (*p* < 0.05 and *p* < 0.01, respectively) compared to the RF mice. Plasma CRE levels were also significantly lower in the RF + PH 10 mice (*p* < 0.05) compared to the RF mice. Tubulointerstitial fibrosis, as evidenced by the expansion of the peritubular spaces, was observed in the RF mice, with the Masson’s trichrome (MT)-stained areas increasing significantly (*p* < 0.01) compared to the control mice. However, this was attenuated in the PH-HPβCD-supplemented mice, which showed significantly lower fibrosis (*p* < 0.01) compared to the RF group, as shown in Figure 3f,g. The experiment was repeated using HPβCD alone instead of the PH-HPβCD solid system to evaluate the effect of HPβCD. The results (Appendix A) indicate a tendency for the plasma IS to increase, whereas BUN and plasma creatinine tended to decrease. However, these results were not significantly different from those of the RF mice. Furthermore, despite a slight decrease, there were no significant differences in the renal fibrosis area of the HPβCD-supplemented mice compared to the RF mice. These suggest that HPβCD alone, at the dose administered, does not result in a significant renoprotective effect.

### 2.4. Treatment with PH-HPβCD Improved Plasma IS and BUN but Not Fibrosis in Moderately Advanced Adenine-Induced CKD Mice

To investigate if administration of PH-HPβCD could improve renal function and tissue fibrosis in moderately advanced CKD, male C57BL/6J mice were fed a diet containing 0.2% w/w adenine for three weeks after acclimatization on a normal diet for a week to create moderately advanced renal failure (protocol shown in Figure 4a). Thereafter, the mice were divided into two groups, the RF and the RF + PH 10 groups. The RF group was fed a normal diet, while the RF + PH 10 group was fed a normal diet supplemented with the formulated PH-HPβCD solid system, equivalent to 10 mg/kg/day of PH, for another four weeks. A control (Ctrl) group was fed a normal diet for the entire study duration. Figure 4a–g shows the effects of treatment with formulated PH on the renal function of moderately advanced adenine-induced CKD mice. The adenine-fed mice showed a decline in body weight and food intake and an increase in water intake. After switching to a normal diet or a PH-HPβCD-supplemented normal diet, both the RF and RF + PH 10 mice slightly regained weight and food intake. However, at the end of the study period, the RF mice showed significantly lower body weight (*p* < 0.01) compared to the control group. The PH-HPβCD-treated mice showed a slightly better weight recovery than the RF mice, albeit not statistically significant. Additionally, no significant differences in the food and water intake between the RF and PH-HPβCD-treated mice were observed at the end of the study (Figure 4b and Appendix A). The plasma IS, BUN, and creatinine were significantly elevated (28.0 ± 8.6 µg/mL, 142.0 ± 1.9 mg/dL, and 0.52 ± 0.04 mg/dL) after the 3-week adenine diet. However, at the end of the study period, there appeared to be a reduction in the levels of these renal markers, with the CRE in particular being markedly reduced (Figure 4c–e). This observed recovery may be attributed to the discontinuation of adenine. Notwithstanding, the renal markers of the RF mice remained significantly higher (*p* < 0.001 for plasma IS and BUN) and (*p* < 0.05 for CRE) compared to the control group. Notably, the PH-HPβCD-treated mice showed better recovery, with significantly lower plasma IS and BUN (*p <* 0.05) and a slightly lower plasma CRE compared to the RF mice. Representative kidney sections (Figure 4f) showed the expansion of the peritubular spaces and the beginning of cyst formation, indicative of tubulointerstitial fibrosis in the RF mice. The treated mice showed less severe cyst formation, with slightly lower total fibrosis area than the RF mice, although this was not statistically significant, as shown in Figure 4g.

### 2.5. PH-HPβCD Did Not Improve Renal Function and Fibrosis in 5/6-Nephrectomized Mice

The effect of PH-HPβCD administration on advanced stages of CKD was also evaluated in a 5/6 nephrectomized CKD mouse model. Male C57BL/6J mice were subjected to 5/6 nephrectomy (Nx) surgery and were allowed to heal and acclimatize over four weeks on a normal diet (protocol shown in Figure 5a). The nephrectomized mice were divided into the RF and the RF + PH 10 groups. The RF group was continued on a normal diet, while the RF + PH 10 group was fed a normal diet supplemented with the formulated PH-HPβCD solid system, equivalent to 10 mg/kg/day of PH, for another four weeks. A control (Ctrl) group (which had not undergone Nx surgery) was fed a normal diet for the entire study duration. One mouse each died in the sixth week of the study from the RF group (day 41) and the RF + PH 10 group (day 37). Figure 5a–g shows the effects of PH-HPβCD administration on the renal function of 5/6 nephrectomized mice. The RF group showed significant weight loss (*p* < 0.05) compared to the control group. There were no significant differences in the body weight, food intake, or water intake of the PH-HPβCD-treated mice and the RF mice (Figure 5b and Appendix A). Moreover, the RF mice showed significantly elevated plasma IS, BUN, and plasma CRE at the end of the study period, which were decreased, albeit not significantly, by PH-HPβCD treatment, as shown in Figure 5c–e. The nephrectomy surgery resulted in vascular degeneration, as seen in the kidney sections of the RF and RF + PH 10 mice (Figure 5f). Extensive cyst formation was observed in the kidney section of the RF mice, which suggests advanced tissue fibrosis. The PH-HPβCD-treated mice also showed peritubular space expansion, with less cyst formation than the RF mice. However, the total fibrosis area was not statistically lower than in the RF mice, as shown in Figure 5g.

## 3. Discussion

The present study explored the potential of blocking the gut bacterial-specific enzyme TIL to suppress indole production in the gut and reduce circulating levels of the uremic toxin IS, thereby delaying or reversing the deterioration of renal function in CKD. This indirect strategy for protecting renal function could provide other therapeutic advantages. For instance, since a microbial-specific enzyme is inhibited in a non-lethal manner in this approach, selective pressure for drug resistance development is expected to decrease [19,29,30]. This is especially relevant considering that metabolic indole is crucial in biofilm formation and bacterial pathogenicity [31,32]. In light of previous studies suggesting that PB and its structurally related compounds may have renoprotective effects, our study aimed to investigate their potential to inhibit TIL activity, suppress IS production, and consequently preserve renal function in CKD.

The results of the in vitro kinetic studies indicate that PB and structurally related compounds possess TIL-inhibitory activities. The efficacy and potency of this inhibition were influenced by the presence of an aromatic ring and the distance (methylene chain length) between the aromatic ring and the carboxylate group, respectively. PH, the most potent among the compounds in the homologous series studied, was shown to block the decomposition of tryptophan into indole. The distance between the phenyl ring and carboxylate group in PH is identical to that of L-bishomotryptophan, which is reported to be the most potent specific inhibitor of TIL [13]. This suggests a structural and spatial complementarity between PH and the binding site of TIL. Additionally, we investigated the effect of PH on sulfotransferase enzyme (SULT) activity in vitro, given that the sulfonation of 3-hydroxyindole to IS by SULT in the liver is a crucial step in the biotransformation of IS [6,33]. Our findings, however, indicate that PH has no significant effect on SULT activity in vitro, as shown in Appendix A.

We used CD complexation to overcome the biopharmaceutical limitations of PH (a poorly soluble viscous oil with an unpleasant odor), as this method provides an avenue to address all the bottlenecks simultaneously [27,34]. Moreover, CDs are biocompatible and have a favorable toxicological profile [35,36]. βCD has been reported to have an optimal cavity size for binding with PH, albeit resulting in only marginal improvements in solubility [28]. Consequently, it is unsuitable for formulating PH, especially considering its low suggested oral threshold (10 mg/kg body weight in humans). HPβCD, which has an identical cavity size and a much higher suggested oral threshold (160 mg/kg body weight), was evaluated. Studies in animals and humans have shown good tolerance to HPβCD, especially when taken orally. The main adverse effects reported at high doses (>1000 mg/kg/day) are diarrhea and reversible hematological changes [37,38]. Our results indicate that HPβCD is more efficient in complexing and solubilizing PH, possibly due to its superior solubility and additional interactions between its hydroxypropyl groups and PH [39]. Freeze-drying PH with HPβCD resulted in the formulation of a free-flowing solid system. The DSC curve obtained from the solid system is consistent with the formation of inclusion complexes. This is apparent from the absence of the distinct sharp endothermic peak at 149 °C, which corresponds to the boiling point of PH, compared to the physical mixture [40]. This complex formation is likely responsible for the significant increase in the dissolution rate and efficiency and masking of the unpleasant odor of PH [41,42]. It is noteworthy that the presence of HPβCD did not offset the in vitro TIL-inhibitory activity of PH despite a reduction in activity. Thus, formulating PH with HPβCD allowed for evaluating its suppression of indole production and potential for renoprotection in vivo.

To determine whether the decrease in indole production observed in vitro would translate into reduced circulating IS and contribute to the slowing or reversal of renal dysfunction in CKD, in vivo experiments were conducted using an adenine-induced CKD mouse model. Adenine is rapidly metabolized to 2,8-dihydroxyadenine after ingestion. This forms crystals in the epithelia of proximal renal tubules, causing renal failure [43]. The RF mice showed reduced food intake, increased water intake, weight loss, and increased plasma IS, BUN, CRE, and tubulointerstitial fibrosis consistent with adenine-induced renal failure [43,44,45]. However, concomitant supplementation with PH-HPβCD attenuated the deterioration of these renal failure markers in a dose-dependent manner. In contrast, supplementation with HPβCD alone produced no significant renoprotective effect. In a previous study, HPβCD reduced circulating IS in nephrectomized rats by adsorbing indole in the gut [46]. However, the dose of HPβCD used in that study (250 mg/kg) was significantly higher than the HPβCD content of the PH-HPβCD solid system employed in the present study (~97 mg/kg). It is conceivable that the HPβCD doses in the current study did not significantly adsorb indole. Instead, HPβCD primarily acts as a carrier for PH, the release of which is expected to occur through simple dilution and displacement by endogenous substances such as bile acids in the gut [35]. Importantly, the HPβCD content of the administered doses of the PH-HPβCD solid system did not elicit diarrhea or loose stools. Thus, our findings suggest that the PH-mediated suppression of indole production, through its inhibition of TIL in the gut, is at least partly responsible for the observed reduction in circulating IS and attenuation of renal failure in the PH-HPβCD supplemented mice.

The renal damage induced by the low adenine diet (0.2% w/w) in mice is slow and progressive, mirroring the progression of CKD in humans. Consequently, this model also provides an avenue for testing potential interventions for reversal [45]. Therefore, we assessed the efficacy of PH-HPβCD to reverse moderate renal failure by administering it to mice fed an adenine diet for three weeks instead of the typical six-week adenine feeding regimen that models advanced renal failure [19,20,44]. PH-HPβCD treatment reduced plasma IS and BUN and showed a tendency to reduce plasma CRE and renal tissue fibrosis. In 5/6 nephrectomized mice treated with PH-HPβCD, there was a tendency to reduce plasma IS, BUN, and plasma CRE. However, no significant improvements in these renal failure markers or tissue fibrosis were observed. Notably, in both the prophylactic and treatment regimens, PH-HPβCD reduced plasma IS levels in a manner that positively correlated with the observed renoprotective effects. These findings support the assertion that PH-HPβCD can decrease circulating IS levels, resulting in significant renal protection in CKD.

However, the extent to which the inhibition of TIL in the gut and suppression of IS production contributes to the observed renoprotective effects remains unclear, as other mechanisms may be concurrently at play. It is plausible that other uremic toxins had been simultaneously reduced, augmenting the observed results. For instance, tyrosine phenol lyase (EC 4.1.99.2), which is involved in phenyl sulfate production, shows high-sequence homology with TIL and may have been partially inhibited by PH [19]. Additionally, PH may have a more direct protective effect on kidney tissues, such as an anti-inflammatory or ER stress inhibitory effect on kidney cells. The formulation could also alter or modify the gut microbiota composition and may have contributed to the observed renoprotective effects [23,24,47]. Further research is required to establish the involvement of these possible mechanisms. Moreover, while no overt adverse effects of administering PH-HPβCD were observed during the study, toxicological studies are necessary to understand its potential impact fully. These considerations would constitute the focus of future studies.

## 4. Conclusions

The present study demonstrates that while advanced CKD was irreversible, prophylactic supplementation or treatment with PH-HPβCD can slow the progression of renal failure in mice with mild to moderate adenine-induced CKD, at least partly by inhibiting TIL in the gut, which in turn reduces the levels of circulating IS. These findings have implications for managing CKD, particularly in early-stage disease.

## 5. Materials and Methods

### 5.1. Materials

PH (>97.2%) was purchased from Alfa Aesar (Heysham, UK), whereas PB as sodium salt (>99.5%), PV (>99.0%), and PC (>98.0%) were obtained from Tokyo Chemical Industry Co., Ltd. (Tokyo, Japan). HA (>98.0%) was sourced from Wako Pure Chemical Industries Ltd. (Osaka, Japan). HPβCD (degree of substitution, 4.9) was obtained from Nihon Shokuhin Kako Co., Ltd. (Tokyo, Japan), whereas βCD (>98.0%) and BA (>95.0%) were purchased from Nacalai Tesque Inc. (Kyoto, Japan). All chemicals were sourced commercially, of the highest analytical grade, and used as received without further purification.

### 5.2. Inhibition of Tryptophan Indole Lyase Activity In Vitro

The assay was conducted in a cuvette (1 cm optical path length) using a UV–vis spectrophotometer equipped with a temperature controller (Model V-550, ETC-505T, Jasco Corp., Tokyo, Japan) at 37 ± 0.5 °C, as previously reported [15,19]. The reaction mixture consisted of 0.005% BSA (Sigma-Aldrich Co., St. Louis, MO, USA), 1 mM reduced glutathione (Sigma), 100 μM pyridoxal phosphate (Cayman Chemical Co., Ann Arbor, MI, USA), 100 μM NADH (Sigma), 3 U/mL lactate dehydrogenase (Sigma), and 10 U/mL apotryptophanase (Sigma) in 0.15 M potassium phosphate buffer (pH 8.0). After incubating for 3 min with various known concentrations of PB or structural homologues (PV, PC, PH, BA, and HA) in 0.15 M potassium phosphate buffer (pH 8.0), the enzymatic reaction was initiated by adding L-tryptophan (300 μM final, Nacalai Tesque). The decrease in optical density at 340 nm was monitored for 5 min at 1 s intervals and used to calculate the TIL reaction velocity. Additionally, the time course of tryptophan decomposition and indole formation in the presence of PH was monitored by terminating the reaction after predetermined reaction times using ice-cold methanol. After centrifugation at 3000× *g* for 10 min, 10 μL of the supernatant was injected into an HPLC system to determine the content of tryptophan and indole. The HPLC conditions were identical to those for the IS measurement (described in Section 5.4.6). All measurements were done in triplicate, at least. The retention times for tryptophan and indole were 2.4 min and 12.8 min, respectively. Standard curves for L-tryptophan (Nacalai Tesque) and indole (Nacalai Tesque) in methanol were used as references.

### 5.3. Preparation and Characterization of PH-HPβCD Solid System

#### 5.3.1. Phase Solubility Studies

Phase solubility studies were performed, as previously reported [48,49]. Briefly, excess PH was added to 1 mL solutions (0 to 3 mM, pH 2.1) of βCD or HPβCD in screw-cap tubes. After shaking at 120 rpm and 25 °C for 72 h (Multi Shaker MMS-3020, FMC-1000; Eyela Co., Ltd., Tokyo, Japan), the suspensions were filtered through 0.2 µm membrane filters (Minisart RC 4, Sartorius Stedim Lab Ltd., Stonehouse, UK). The solubility of PH was measured by HPLC, using a JASCO HPLC system (Jasco Corp., Tokyo, Japan) [49]. The data were used to construct solubility diagrams, and the complexation efficiency (CE) of the CD complexes was calculated using the slope of the initial linear portion of the solubility diagrams according to the equation CE = slope/1 − slope [39].

#### 5.3.2. Preparation of PH-HPβCD Solid System

Equimolar amounts (2 mmol) of PH (0.41 g) and HPβCD (2.84 g in 100 mL water) were mixed and continuously stirred (120 rpm, 25 °C, seven days). The mixture was filtered (0.2 µm), and the resulting clear solution was lyophilized (Eyela FDU-1200, Tokyo Rikakikai Co., Ltd., Tokyo, Japan) to obtain a white powder [42]. The PH content of the prepared solid system was determined by HPLC and then stored in a digitally controlled desiccator (AS ONE Corp., Osaka, Japan).

#### 5.3.3. Characterization of PH-HPβCD Solid System

The surface morphology and particle structure of the PH-HPβCD solid system were examined using a scanning electron microscope (Miniscope TM 3000, Hitachi High-Tech. Corp., Tokyo, Japan) operating at 5.0 kV. The samples were fixed onto an SEM stub using double-sided carbon sticky tape and imaged at a magnification of 200 [42].

The thermal behavior of the prepared PH-HPβCD solid system, HPβCD alone, and a physical mixture of HPβCD and PH (content identical to the prepared solid system) was studied using a differential scanning calorimeter (Model 8240E1 Thermoplus, Rigaku Corp., Tokyo, Japan). Five milligrams of each sample was sealed in a 40 µL aluminum pan, with an identical pan containing Al_2_O_3_ used as a reference. The samples were heated from 25 to 250 °C at a 10 °C/min heating rate under a 200 mL/min nitrogen purge.

The dissolution rate of 200 μg PH or its equivalent amount of the PH-HPβCD solid system was measured using a modified beaker method in 100 mL of Japanese Pharmacopoeia (JP) 1st dissolution fluid (prepared by dissolving 2.0 g of sodium chloride in 7.0 mL of hydrochloric acid and water to make 1 L with pH 1.2) in a stoppered glass vial. The dissolution medium was maintained at 37 ± 0.5 °C by means of a water bath and was stirred with a magnetic bar (29 mm × 6 mm) at 75 rpm. The 200 μg of PH would achieve a maximum concentration of 2.0 μg/mL in the 100 mL dissolution medium. This met sink conditions because the saturated concentration of PH in the dissolution medium was measured to be 2.7 ± 0.11 μg/mL in preliminary solubility tests. At predetermined time intervals, 0.5 mL aliquots were withdrawn from the beaker and transferred into HPLC vials for assay. To maintain the total liquid volume in the setup, 0.5 mL of fresh dissolution medium kept at 37 ± 0.5 °C was immediately added to the medium after each sampling. The measurement was conducted for 60 min. The dissolution rate measurement was performed in triplicate. The dissolution profiles were evaluated by the percentage of drug dissolved at 10 min (DP_10min_) and the dissolution efficiency at 60 min (DE_60min_), calculated from the area under the dissolution curve [42,50].

### 5.4. Animal Experiments

#### 5.4.1. Animal Care

Healthy male C57BL/6J mice, five weeks old (weight 18–20 g), and male C57BL/6J mice that had been subjected to a 5/6 nephrectomy were obtained from Japan SLC Inc. (Shizuoka, Japan). The mice were kept under standard conditions (12 h light/dark schedule, 23 ± 1 °C) with free access to a regular diet for breeding (CE-02 powder, Clea Japan Inc., Tokyo, Japan) and drinking water ad libitum before the start of experiments. All the animal experiments were approved and performed according to protocols reviewed and approved by the Animal Care and Use Committee of Sojo University (Approval No. 2022-P-022).

#### 5.4.2. Prophylactic Supplementation with PH-HPβCD in Adenine-Induced CKD Mice

After acclimatizing for a week, healthy mice were randomly divided into control (Ctrl), renal failure (RF), and two renal failure with prophylactic PH-HPβCD supplementation groups, i.e., RF + PH 5 and RF + PH 10. The Ctrl group was maintained on a regular (normal) diet; the RF group was fed a 0.2% w/w adenine diet, while the RF + PH groups were fed 0.2% w/w adenine diets supplemented with PH-HPβCD solid system (≈5 or 10 mg/kg/day of PH) for three weeks. The doses of PH were chosen based on the findings of a preliminary study using a dose of 1 mg/kg/day of PH. Additionally, the daily intake of HPβCD was considered to minimize adverse effects like diarrhea. The chosen doses ensured an HPβCD dose not exceeding 100 mg/kg/day throughout the study.

#### 5.4.3. Treatment with PH-HPβCD in Moderately Advanced Adenine-Induced CKD Mice

After acclimatizing for a week, healthy mice were randomly divided into Ctrl, RF, and RF + PH 10 groups. The Ctrl group was maintained on a regular (normal) diet for seven weeks, while the RF and RF + PH 10 groups were fed a 0.2% w/w adenine diet for three weeks. Afterwards, they were switched to a normal diet or a normal diet supplemented with PH-HPβCD solid system (≈10 mg/kg/day of PH), respectively, for another four weeks.

#### 5.4.4. Treatment with PH-HPβCD in 5/6-Nephrectomized Mice

The 5/6 nephrectomy was performed on five-week-old healthy male C57BL/6J mice (weight 18–20 g), involving the resection of two-thirds of the left kidney, followed by the complete excision of the right kidney one week later. Briefly, the left kidney was removed from the abdominal cavity under isoflurane anesthesia, and blood flow in the renal artery and vein was halted using a vascular clamp. The upper and lower thirds of the kidney were then excised with a slicer, and Surgicel (Johnson & Johnson MedTech, Raritan, NJ, USA) was applied for hemostasis. Following removal of the clamp, the kidney was restored in the abdominal cavity and sutured, with subsequent subcutaneous administration of enrofloxacin (5 mg/kg) and buprenorphine (0.1 mg/kg). Analgesic administration was repeated the following day. One week later, total excision of the right kidney was carried out, and post-operative analgesics were administered until the day after the surgery. After the surgery, the mice were allowed to heal and acclimatize on a normal diet for four weeks. The nephrectomized mice were then randomly divided into two groups, i.e., the RF and the RF + PH 10 groups. The RF group was maintained on a normal diet, while the RF + PH 10 group was switched to a normal diet supplemented with PH-HPβCD solid system (≈10 mg/kg/day of PH) for another four weeks. A control group of healthy mice without nephrectomy surgery was maintained on a normal diet for the entire study duration.

All the animals were supplied with the respective feed and tap water ad libitum throughout the study. At the end of the study periods, blood samples were collected under isoflurane anesthesia (FUJIFILM, Wako Pure Chemicals Corp., Osaka, Japan), and plasma from the samples was immediately frozen at −80 °C until analyzed. The mice were euthanized by exsanguination, and the kidneys were excised and fixed in 10% phosphate-buffered formalin (FUJIFILM) for histological analyses.

#### 5.4.5. Histological Analysis

The mouse kidneys were fixed in 10% phosphate-buffered formalin and embedded in paraffin. Kidney sections were stained with Masson’s trichrome (MT) and imaged using a BZ-X700 microscope (Keyence Corp., Osaka, Japan). The tubulointerstitial fibrosis area was quantified using the National Institute of Health Image J/Fiji v2.9.0.

#### 5.4.6. Measurement of IS, BUN, and CRE in Mouse Plasma

Mouse plasma IS was measured according to a previous report with some modifications [51]. First, a 20 μL aliquot of mouse plasma was deprotonated by adding 140 μL acetonitrile. Then, after centrifugation at 3000× *g* for 10 min, 20 μL of the supernatant was injected into an HPLC system consisting of a PU-2089 Plus intelligent pump and an FP-2020 Plus fluorescence spectrophotometer (Jasco Corp., Tokyo, Japan). An Intersil ODS-3 column (250 mm × 4.6 mm, 5 µm, GL Sciences Inc., Tokyo, Japan) was used as stationary phase and was maintained at 40 °C. The mobile phase consisted of acetate buffer (0.2 M, pH 4.5)/acetonitrile programmed at a flow rate of 1.0 mL/min. IS was detected using a fluorescence monitor with excitation and emission wavelengths of 280 nm and 370 nm, respectively, with a retention time of 4.2 min. A standard curve for IS (Nacalai Tesque) in methanol was used as the reference.

BUN and plasma CRE were measured using an automatic analyzer (Hitachi 7180, Hitachi High-Tech. Corp., Tokyo, Japan).

### 5.5. Statistical Analysis

Statistical significance was evaluated by the two-tailed Student’s *t*-test for comparison between two mean values and by a one-way ANOVA, followed by the Bonferroni correction for comparison between multiple groups. A *p*-value < 0.05 was considered significant.

## Figures and Tables

**Figure 1 toxins-16-00316-f001:**
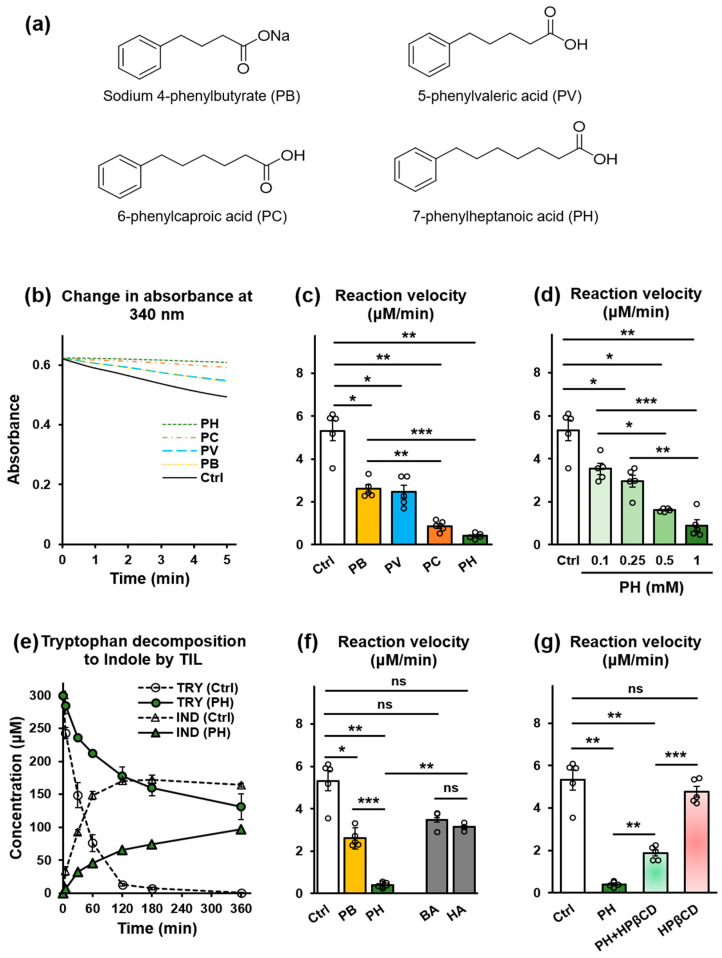
Effects of PB and structurally related compounds on TIL activity in vitro. (**a**) Chemical structures of PB and related compounds. (**b**) Change in absorbance of reaction medium at 340 nm. (**c**) TIL activity in the presence of PB or structurally related compounds (2 mM). (**d**) TIL activity in the presence of increasing concentrations of PH. (**e**) Time course of the decomposition of tryptophan (TRY) to indole (IND) by TIL in the presence of PH (2 mM). (**f**) Effect of the aromatic ring on the TIL-inhibitory activity of PB and PH (2 mM). (**g**) Effect of HPβCD on the TIL-inhibitory activity of PH. Values expressed as mean ± SE (*n* = 5; except for d, where *n* = 3). One-way ANOVA, followed by Bonferroni correction. * *p* < 0.05, ** *p* < 0.01, *** *p* < 0.001, ns, no significant difference.

**Figure 2 toxins-16-00316-f002:**
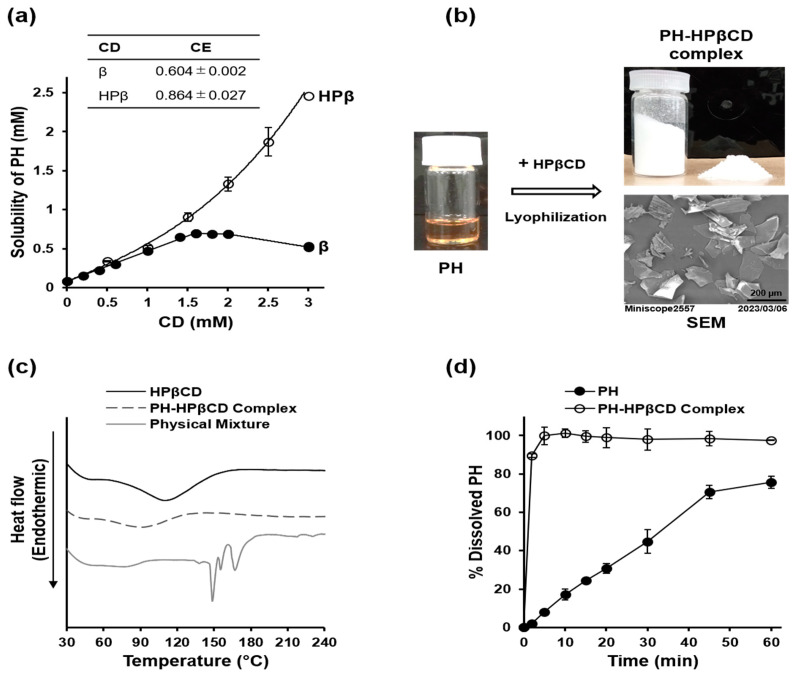
Preparation and characterization of PH-HPβCD solid system. (**a**) Phase solubility diagrams of PH-βCD and PH-HPβCD systems in 0.1 M phosphate buffer (pH 2.1) at 25 °C. Inset is the complexation efficiency (CE) of the systems. (**b**) Appearance of PH (oil) before, and PH-HPβCD solid system (white powder) after lyophilization with HPβCD, and SEM image of the PH-HPβCD solid system. (**c**) DSC curves of HPβCD, PH-HPβCD solid system, and PH-HPβCD physical mixture. (**d**) Dissolution profiles of PH alone and PH-HPβCD solid system in JP 1st dissolution fluid (pH 1.2) at 37 ± 0.5 °C. Each point represents the mean ± SE (*n* = 3) for a and d.

**Figure 3 toxins-16-00316-f003:**
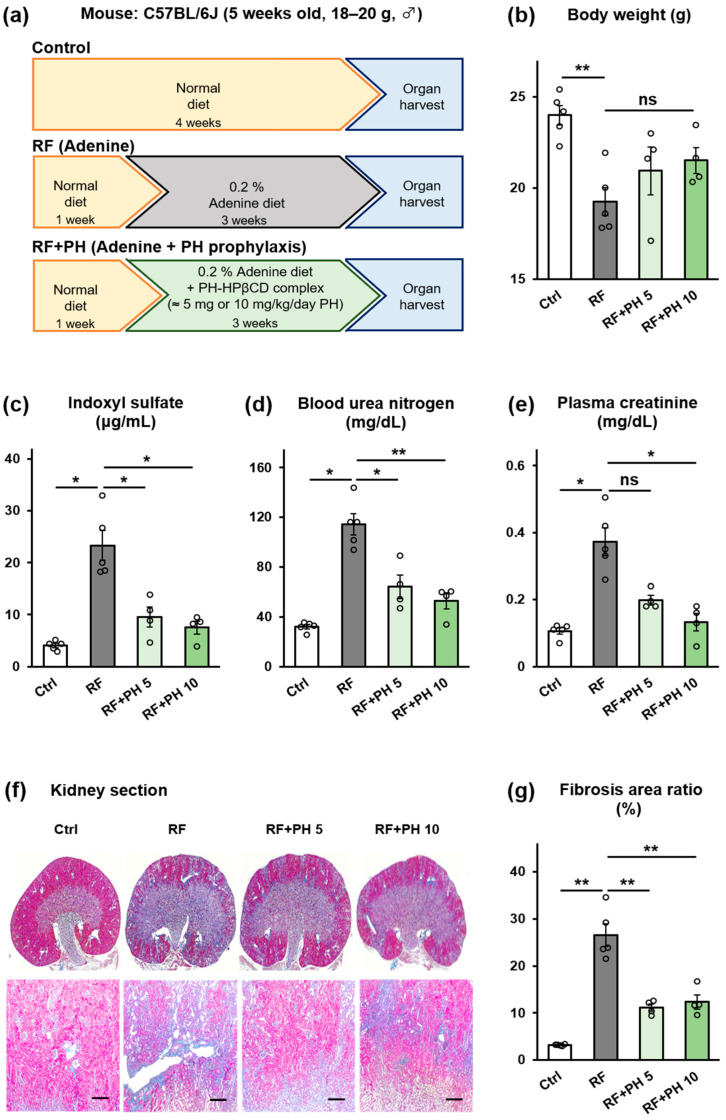
Effect of prophylactic supplementation with PH-HPβCD on the renal function of adenine-induced CKD mice. (**a**) Schematic overview of study protocol. (**b**) Body weight. (**c**) Plasma IS. (**d**) BUN. (**e**) Plasma CRE. (**f**) Representative micrographs of Masson’s trichrome-stained kidney sections; scale bar, 200 μm. (**g**) Quantitative analysis of tubular fibrosis area. The actual doses of PH for the RF + PH 5 and RF + PH 10 groups, based on food intake, were 5.35 ± 0.19 mg/kg/day and 11.96 ± 0.53 mg/kg/day, respectively. Values expressed as mean ± SE (*n* = 4–5). One-way ANOVA, followed by Bonferroni correction. * *p* < 0.05, ** *p* < 0.01, ns, no significant difference.

**Figure 4 toxins-16-00316-f004:**
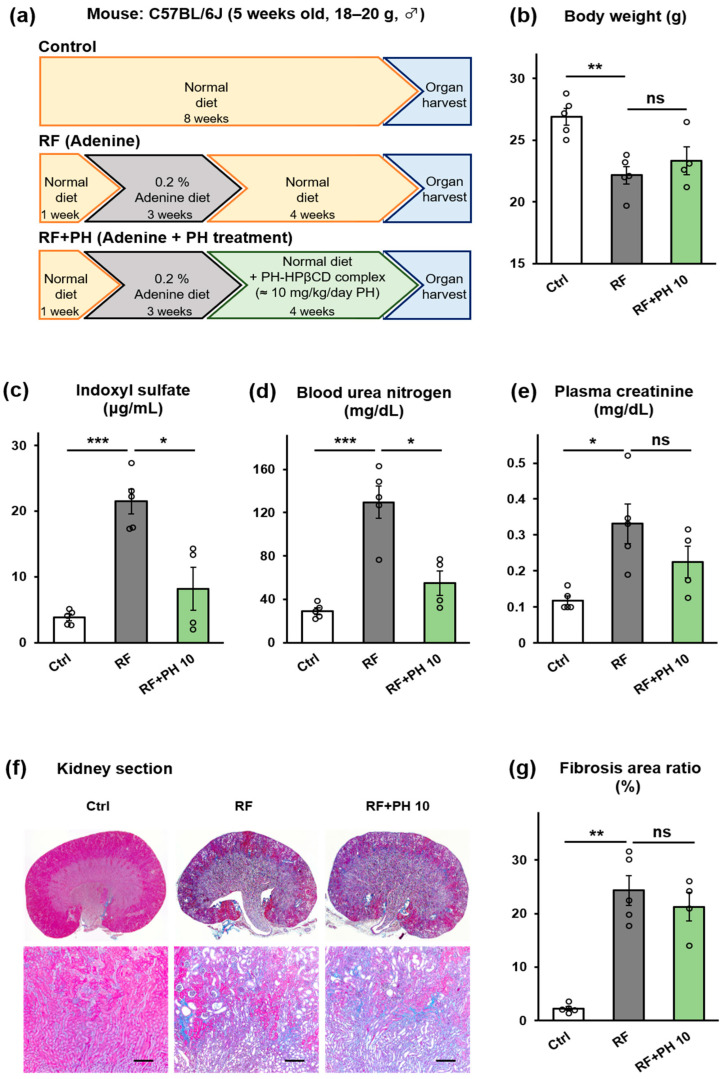
Effect of treatment with PH-HPβCD on the renal function of moderately advanced adenine-induced CKD mice. (**a**) Schematic overview of study protocol. (**b**) Body weight. (**c**) Plasma IS. (**d**) BUN. (**e**) Plasma CRE. (**f**) Representative micrographs of Masson’s trichrome-stained kidney sections; scale bar, 200 μm. (**g**) Quantitative analysis of tubular fibrosis area. The actual dose of PH for the RF + PH 10 group, based on food intake, was 10.04 ± 0.34 mg/kg/day. Values expressed as mean ± SE (*n* = 4–5). One-way ANOVA, followed by Bonferroni correction. * *p* < 0.05, ** *p* < 0.01, *** *p* < 0.001, ns, no significant difference.

**Figure 5 toxins-16-00316-f005:**
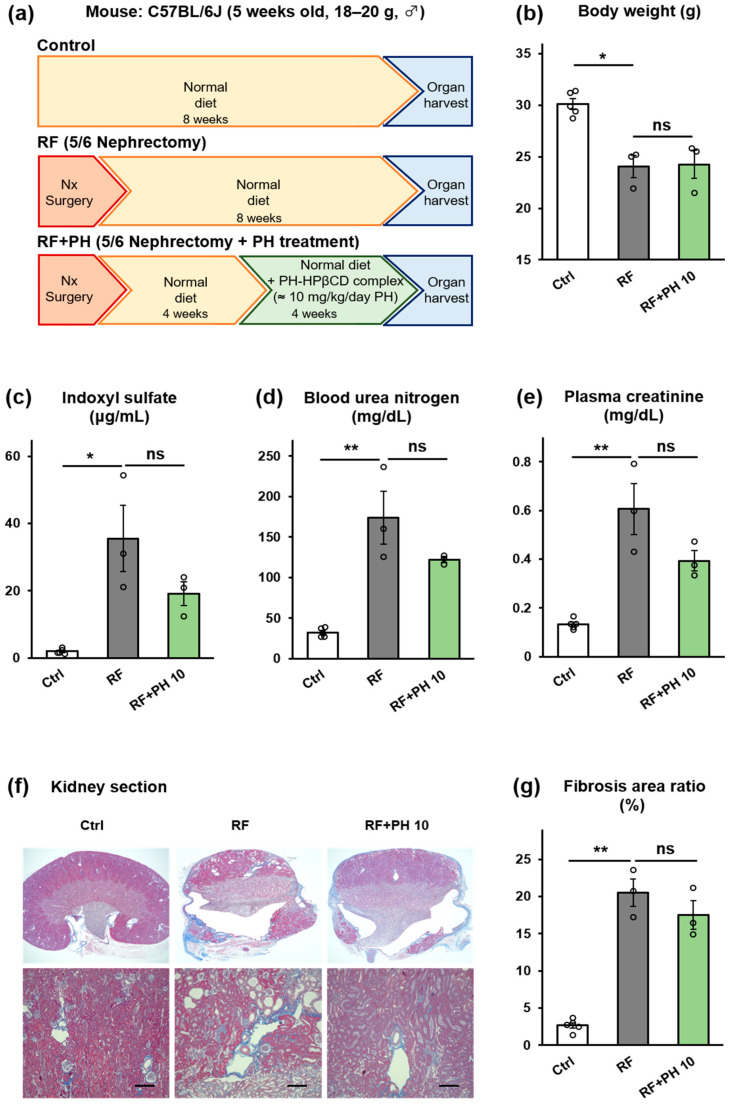
Effect of treatment with PH-HPβCD on the renal function of 5/6 nephrectomized mice. (**a**) Schematic overview of study protocol. (**b**) Body weight. (**c**) Plasma IS. (**d**) BUN. (**e**) Plasma CRE. (**f**) Representative micrographs of Masson’s trichrome-stained kidney sections; scale bar, 200 μm. (**g**) Quantitative analysis of tubular fibrosis area. The actual dose of PH for the RF + PH 10 group, based on food intake, was 10.31 ± 0.44 mg/kg/day. Values expressed as mean ± SE (*n* = 3–5). One-way ANOVA, followed by Bonferroni correction. * *p* < 0.05, ** *p* < 0.01, ns, no significant difference.

## Data Availability

The original contributions presented in the study are included in the article/Appendix A. Further inquiries can be directed to the corresponding author.

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
