# Peer review of "7-Phenylheptanoic Acid-Hydroxypropyl β-Cyclodextrin Complex Slows the Progression of Renal Failure in Adenine-Induced Chronic Kidney Disease Mice"

_toxins, 2024, doi:10.3390/toxins16070316_

Round 1

Reviewer 1 Report

Comments and Suggestions for Authors

In this manuscript, the authors demonstrated the role of 7-Phenylheptanoic acid- Hydroxypropyl β-Cyclodextrin treatment in vitro and in both adenine-induced nephropathy and 5/6 nephrotomy mice. PH presents the inhibition of tryptophan indole lyase activity to decrease indoxyl sulfate in serum levels. The story sounds interesting. However, some issues should be addressed as below:

1. In Fig. 1, the chemical structure of PB, PV, PC, and PH should help readers follow the later description.

2. In fig. 3e, and 4e, why does the marker BUN significantly elevate, but plasma creatinine concentration slightly increases in the normal range of the AD treatment group? The serum history parameters of C57BL/6J mice could be offered to help readers.

3. In fig. 3f, renal section (RF+PH5) mismatches with the statistical data. The authors should confirm.

4.In Fig. 5f, the pathology of kidney section presents vascuolar degeneration areas in 5/6 nephrectomy and 5/6 nephrectomy+PH? How do the authors surgically administrate this model that should be described clearly in the methods and explain the pathological change?

5. The consideration of the selected dose should be discussed

6. HPβCD may present minor side effects such as diarrhea and systemic hemodynamics. (PMID: 23978386; PMID: 16018907……) The authors could discuss the adverse impact of HPβCD in the discussion.

7. The grammar errors and typos should be proofread.

Comments on the Quality of English Language

The English writing is fluent and easy to read. However, it is still suggested to proofread by a native English speaker. 

Author Response

We are grateful to the editor and the reviewers for their helpful comments regarding our manuscript, “7-phenylheptanoic acid-hydroxypropyl β-cyclodextrin complex slows the progression of renal failure in adenine-induced chronic kidney disease mice" (Manuscript ID: toxins-3045770). 

Reviewer 1

Comments and Suggestions for Authors:

In this manuscript, the authors demonstrated the role of 7-Phenylheptanoic acid- Hydroxypropyl β-Cyclodextrin treatment in vitro and in both adenine-induced nephropathy and 5/6 nephrectomy mice. PH presents the inhibition of tryptophan indole lyase activity to decrease indoxyl sulfate in serum levels. The story sounds interesting. However, some issues should be addressed as below:

Comment 1: In Fig. 1, the chemical structure of PB, PV, PC, and PH should help readers follow the later description.

Response: Thank you very much for the suggestion. We have revised Figure 1 to include the chemical structures of PB, PV, PC, and PH.

Comment 2: In fig. 3e, and 4e, why does the marker BUN significantly elevate, but plasma creatinine concentration slightly increases in the normal range of the AD treatment group? The serum history parameters of C57BL/6J mice could be offered to help reader.

Response: Thank you for pointing this out. In Figure 3, both the BUN and plasma creatinine are elevated about 4-fold in the adenine-fed mice. However, in Figure 4, the plasma creatinine shows only a slight increase, though BUN was significantly elevated. We considered the serum history parameters of the mice and refined the statistical analysis of the data. The serum history indicates that the BUN and plasma creatine levels were significantly elevated (4 to 5-fold) after the 3-week adenine diet. However, the mice appeared to partly recover after discontinuing the adenine diet, with the plasma creatinine levels showing a more marked decrease than the BUN. However, both the BUN and plasma creatinine of the adenine-fed mice remained significantly higher than control. This phenomenon was also reported by Kumakura et al. (Ref.  44) even after a 6-week adenine diet. Our results, however, suggest that treatment with the formulated PH could hasten this recovery. As suggested by the reviewer, we have included information on the BUN and plasma creatinine history in the Results section of the revised manuscript as follows:

[Original sentences]

“However, the treated mice showed significantly lower plasma IS and BUN (p < 0.05) compared to the RF mice (Figure 4c, d). No significant improvement in plasma CRE levels was observed, although there was a tendency to decrease, as shown in Figure 4e.”

[New sentences in revised manuscript]

“The plasma IS, BUN, and creatinine were significantly elevated (28.0±8.6 µg/mL, 142.0±1.9 mg/dL, and 0.52±0.04 mg/dL) after the 3-week adenine diet. However, at the end of the study period, there appeared to be a reduction in the levels of these renal markers, with the CRE in particular being markedly reduced (Figure 4c-e). This observed recovery may be attributed to the discontinuation of adenine. Notwithstanding, the renal markers of the RF mice remained significantly higher (p < 0.001 for plasma IS and BUN) and (p < 0.05 for CRE) compared to the control group. Notably, the PH-HPβCD-treated mice showed better recovery, with significantly lower plasma IS and BUN (p < 0.05) and a slightly lower plasma CRE compared to the RF mice.” (Page 8; Line 208-216).

Comment 3: In fig. 3f, renal section (RF+PH5) mismatches with the statistical data. The authors should confirm.

Response: We agree with the reviewer that the statistical data and the kidney section appear to be mismatched. However, after carefully rechecking the kidney sections and reanalyzing the data for all the mice, we confirm the results as presented.

Comment 4: In Fig. 5f, the pathology of kidney section presents vascuolar degeneration areas in 5/6 nephrectomy and 5/6 nephrectomy+PH? How do the authors surgically administrate this model that should be described clearly in the methods and explain the pathological change?

Response: We apologize for not describing the nephrectomy surgery procedure. We have included a clear description of the procedure in the Materials and Methods section and briefly described the pathological changes in the Results section of the revised manuscript, as suggested by the reviewer, as follows:

[New sentences in revised manuscript]

“5/6 nephrectomy was performed on five-week-old healthy male C57BL/6J mice (weight 18-20 g), involving the resection of two-thirds of the left kidney, followed by the complete excision of the right kidney one week later. Briefly, the left kidney was removed from the abdominal cavity under isoflurane anesthesia, and blood flow in the renal artery and vein was halted using a vascular clamp. The upper and lower thirds of the kidney were then excised with a slicer, and Surgicel (Johnson & Johnson MedTech, Raritan, NJ, USA) was applied for hemostasis. Following removal of the clamp, the kidney was restored in the abdominal cavity and sutured, with subsequent subcutaneous administration of enrofloxacin (5 mg/kg) and buprenorphine (0.1 mg/kg). Analgesic administration was repeated the following day. One week later, total excision of the right kidney was carried out, and post-operative analgesics were administered until the day after the surgery.” (Page 15; Line 471-482).

[New sentences in revised manuscript]

“The nephrectomy surgery resulted in vascular degeneration, as seen in the kidney sections of the RF and RF+PH 10 mice (Figure 5f). Extensive cyst formation was observed in the kidney section of the RF mice, which suggests advanced tissue fibrosis. The PH-HPβCD-treated mice also showed peritubular space expansion, with less cyst formation than the RF mice. However, the total fibrosis area was not statistically lower than in the RF mice, as shown in Figure 5g.” (Page 8; Line 237-243).

Comment 5: The consideration of the selected dose should be discussed.

Response: Thank you for the suggestion. The considerations involved in selecting the doses used in the study have been included in the Materials and Methods section as follows:

[New sentences in revised manuscript]

“The doses of PH were chosen based on the findings of a preliminary study using a dose of 1 mg/kg/day of PH. Additionally, the daily intake of HPβCD was considered to minimize adverse effects like diarrhea. The chosen doses ensured an HPβCD dose not exceeding 100 mg/kg/day throughout the study.” (Page 14; Line 458-461).

Comment 6: HPβCD may present minor side effects such as diarrhea and systemic hemodynamics. (PMID: 23978386; PMID: 16018907……) The authors could discuss the adverse impact of HPβCD in the discussion.

Response: Thank you for the suggestion. The impact of HPβCD has been included in the Discussion section of the revised manuscript as follows:

[Original sentence]

“Thus, HPβCD, with an identical cavity size and a much higher suggested oral threshold (160 mg/kg body weight), was evaluated.”

[New sentences in revised manuscript]

“HPβCD, which has an identical cavity size and a much higher suggested oral threshold (160 mg/kg body weight), was evaluated. Studies in animals and humans have shown good tolerance to HPβCD, especially when taken orally. The main adverse effects reported at high doses (> 1000 mg/kg/day) are diarrhea and reversible hematological changes [37,38].” (Page 11; Line 291-295).

“Importantly, the HPβCD content of the administered doses of the PH-HPβCD solid system did not elicit diarrhea or loose stools.” (Page 12; Line 324-326).

Comment 7: The grammar errors and typos should be proofread.

Response: We apologize for any grammatical errors or typos. The manuscript has been proofread to correct these errors.

Comments on the Quality of English Language: The English writing is fluent and easy to read. However, it is still suggested to proofread by a native English speaker.

Response: Thank you for recommending a proofread. Accordingly, our manuscript has been proofread by a colleague who is proficient in English writing. The following changes have been made in the revised manuscript:

[Original sentence]

“Meanwhile, gut microbial metabolism generates uremic toxins [10–12].”

[New sentence in revised manuscript]

“Meanwhile, gut microbial metabolism generates uremic toxins, thus establishing a vicious cycle [10–12].” (Page 1; Line 43-45).

[Original sentence]

“As nearly all indole present in the gut is derived from the TIL-catalyzed breakdown of dietary tryptophan, and TIL is not expressed in eukaryotes, we hypothesized that inhibiting TIL could decrease the production of indole and lower circulating levels of IS in CKD without any host toxicity [13,19].”

[New sentences in revised manuscript]

“We hypothesized that inhibiting TIL, which is responsible for the breakdown of dietary tryptophan, could reduce indole production and subsequently lower circulating levels of IS in CKD without inducing host toxicity. This is based on the fact that nearly all indole in the gut originates from the TIL-catalyzed degradation of dietary tryptophan and that TIL is not expressed in eukaryotes [13,19].” (Page 2; Line 53-58).

[Original sentences]

“These reports suggest that PB and structurally related compounds could be useful in preserving renal function in CKD. However, the effect of PB and related compounds on uremic toxin, particularly IS production and renal function in CKD, is unknown.”

[New sentences in revised manuscript]

“These reports suggest that PB and its structurally related compounds may have the potential to preserve renal function in CKD. However, their effect on uremic toxin, particularly IS production and renal function in CKD, is unknown.” (Page 2; Line 68-70).

[Original sentence]

“For instance, since in this approach, a microbial-specific enzyme is inhibited in a non-lethal manner, selective pressure for drug resistance development is expected to decrease [19,29,30].”

[New sentence in revised manuscript]

“For instance, since a microbial-specific enzyme is inhibited in a non-lethal manner in this approach, selective pressure for drug resistance development is expected to decrease [19,29,30].” (Page 11; Line 265-267).

[Original sentence]

“However, the results of our study indicate that PH does not affect SULT activity in vitro, as shown in Figure S5.”

[New sentence in revised manuscript]

“Our findings, however, indicate that PH has no significant effect on SULT activity in vitro, as shown in Figure S5.” (Page 11; Line 283-284).

[Original sentence]

“As the kidney damage induced by the low adenine diet (0.2% w/w) in mice is slow and progressive, as in CKD in humans, this model also allows for testing interventions for possible reversal [44].”

[New sentences in revised manuscript]

“The renal damage induced by the low adenine diet (0.2% w/w) in mice is slow and progressive, mirroring the progression of CKD in humans. Consequently, this model also provides an avenue for testing potential interventions for reversal [45].” (Page 12; Line 330-332).

[Original sentence]

“Additionally, a more direct anti-inflammatory or ER stress inhibitory effect of PH on kidney tissue, as well as an alteration or modification of the gut microbiota composition by the formulation, could be involved in the observed renoprotective effects [23,24,46].”

[New sentences in revised manuscript]

“Additionally, PH may have a more direct protective effect on kidney tissues, such as an anti-inflammatory or ER stress inhibitory effect on kidney cells. The formulation could also alter or modify the gut microbiota composition and may have contributed to the observed renoprotective effects [23,24,47].” (Page 12; Line 349-353).

Reviewer 2 Report

Comments and Suggestions for Authors

The aim of this article is to investigate the role of 4-phenylbutyrate (PB) and structurally related compounds in kidney function. The authors explore their effects using both in vitro and in vivo models, including prophylactic, curative, and various chronic kidney disease (CKD) mouse models. The approach is interesting, and the research question is significant. The methodologies employed are innovative and original. However, I have several concerns that need to be addressed:

·  The production methods for 4-phenylbutyrate and structurally related compounds are not clearly described. Further details are needed in the manuscript.

·  Why is there a focus solely on indoxyl sulfate (IS)? Is there any data available on indole-3-acetic acid (IAA) or other indole derivatives?

·  Is there any information on changes in gut microbiota composition?

·  Do you have data on proteinuria? The study mentions fibrosis, but is there information on glomerular structure? Additionally, including other fibrosis markers, such as Col1A1 and TGF-β, could increase the sensitivity of your findings.

·  The statistical analysis needs refinement. When comparing groups, ANOVA with post hoc analysis is more appropriate.

·  The number of mice in each experiment is quite modest (n=5 maximum) given the variability in kidney function alterations by all models. This warrants a cautious interpretation of the results, particularly concerning the lack of effect.

·  The evidence for causality in nephroprotection would be stronger if in vitro cell models were also included.

·  The absence of specific effects of HCBCD must be more thoroughly investigated. Additionally, there is a trend towards a decrease in BUN and creatinine levels. Is there any histological analysis available to support these findings?

·  A similar analysis using PH alone (e.g., via oral gavage) would be interesting to clearly delineate the specific nephroprotective effects of this metabolite.

Author Response

We are grateful to the editor and the reviewers for their helpful comments regarding our manuscript, “7-phenylheptanoic acid-hydroxypropyl β-cyclodextrin complex slows the progression of renal failure in adenine-induced chronic kidney disease mice" (Manuscript ID: toxins-3045770). 

Reviewer 2

Comments and Suggestions for Authors: The aim of this article is to investigate the role of 4-phenylbutyrate (PB) and structurally related compounds in kidney function. The authors explore their effects using both in vitro and in vivo models, including prophylactic, curative, and various chronic kidney disease (CKD) mouse models. The approach is interesting, and the research question is significant. The methodologies employed are innovative and original. However, I have several concerns that need to be addressed:

Comment 1: The production methods for 4-phenylbutyrate and structurally related compounds are not clearly described. Further details are needed in the manuscript.

Response: We appreciate the reviewer’s comments. The 4-phenylbutyrate and structurally related compounds used in the study were obtained from commercial sources and were of high purity (> 97%). They were used as received without any further purification.

The following modifications have been made in the revised manuscript to reflect these points:

[Original sentences]

“PH was purchased from Alfa Aesar (Heysham, UK)., whereas PB, PV, and PC were obtained from Tokyo Chemical Industry Co., Ltd. (Tokyo, Japan). HA was sourced from Wako Pure Chemical Industries Ltd. (Osaka, Japan). HPβCD (degree of substitution, 4.9) was obtained from Nihon Shokuhin Kako Co. Ltd. (Tokyo, Japan), whereas βCD and BA were purchased from Nacalai Tesque Inc. (Kyoto, Japan). All other chemicals were obtained from commercial sources and were of the highest analytical grade.”

[New sentence in revised manuscript]

“PH (>97.2%) was purchased from Alfa Aesar (Heysham, UK)., whereas PB as sodium salt (>99.5%), PV (>99.0%), and PC (>98.0%) were obtained from Tokyo Chemical Industry Co., Ltd. (Tokyo, Japan). HA (>98.0%) was sourced from Wako Pure Chemical Industries Ltd. (Osaka, Japan). HPβCD (degree of substitution, 4.9) was obtained from Nihon Shokuhin Kako Co. Ltd. (Tokyo, Japan), whereas βCD (>98.0%) and BA (>95.0%) were purchased from Nacalai Tesque Inc. (Kyoto, Japan). All chemicals were sourced commercially, of the highest analytical grade, and used as received without further purification.” (Page 13; Line 366-372).

Comment 2: Why is there a focus solely on indoxyl sulfate (IS)? Is there any data available on indole-3-acetic acid (IAA) or other indole derivatives?

Response:  We appreciate the reviewer’s request for data on additional indole-derived uremic toxins, such as indole-3-acetic acid (IAA). Given that this was the first study on the role of PB and its homologs in blocking TIL activity, we focused on monitoring the levels of the extensively studied uremic toxin, indoxyl sulfate. Therefore, we currently have no data on other indole derivatives. However, based on our findings, we have broadened our research to explore additional mechanisms contributing to the renoprotective effects, including monitoring other uremic toxins.

Comment 3: Is there any information on changes in gut microbiota composition?

Response: We appreciate the reviewer’s request for information regarding changes in gut microbiota composition. We did not collect data on these changes in this study; however, this is the subject of our currently ongoing investigation.

Comment 4: Do you have data on proteinuria? The study mentions fibrosis, but is there information on glomerular structure? Additionally, including other fibrosis markers, such as Col1A1 and TGF-β, could increase the sensitivity of your findings.

Response: We do not have data on proteinuria or other fibrosis markers. We agree that this information could increase the sensitivity of our findings. Therefore, as the reviewer suggested, we will obtain these additional datasets in our ongoing studies.

Comment 5: The statistical analysis needs refinement. When comparing groups, ANOVA with post hoc analysis is more appropriate.

Response: We appreciate the reviewer’s suggestion and have refined the statistical analysis accordingly.

Comment 6: The number of mice in each experiment is quite modest (n=5 maximum) given the variability in kidney function alterations by all models. This warrants a cautious interpretation of the results, particularly concerning the lack of effect.

Response: We have considered the reviewer’s comments about interpreting the results in light of the limited sample sizes. Firstly, we have refined the statistical analysis per the reviewer’s earlier suggestion. Additionally, we have revised the wording of statements that interpret the results, particularly where there seems to be a lack of effect. The modifications in the revised manuscript reflecting a more cautious interpretation of the results are as follows:

[Original sentences]

“The experiment was repeated using HPβCD alone instead of the PH-HPβCD solid system to evaluate the effect of HPβCD. The results showed no significant differences in plasma IS, BUN, and plasma CRE compared to the RF group (Figure S2a-c).”

[New sentence in revised manuscript]

“The experiment was repeated using HPβCD alone instead of the PH-HPβCD solid system to evaluate the effect of HPβCD. The results (Figure S2a-e) indicate a tendency for the plasma IS to increase, whereas BUN and plasma creatinine tended to decrease. However, these were not significantly different from the RF mice. Furthermore, despite a slight decrease, there were no significant differences in the renal fibrosis area of the HPβCD-supplemented mice compared to the RF mice. These suggest that HPβCD alone, at the dose administered, does not result in a significant renoprotective effect.” (Page 6; Line 174-180).

[Original sentence]

“There were no significant differences in the body weight, food intake, or water intake of the PH-HPβCD-treated and the RF mice (Figure 4b, S3a, b).

[New sentences in revised manuscript]

“However, at the end of the study period, the RF mice showed significantly lower body weight (p < 0.01) compared to the control group. The PH-HPβCD-treated mice showed a slightly better weight recovery than the RF mice, albeit not statistically significant. Additionally, no significant differences in the food and water intake between the RF and PH-HPβCD-treated mice were observed at the end of the study (Figure 4b, S3a, b).” (Page 8; Line 203-208).

[Original sentences]

“However, the treated mice showed significantly lower plasma IS and BUN (p < 0.05) compared to the RF mice (Figure 4c, d). No significant improvement in plasma CRE levels was observed, although there was a tendency to decrease, as shown in Figure 4e. Adenine-induced renal fibrosis was observed in the RF and the RF+PH 10 groups, with no significant differences, as shown in Figure 4f, g.).”

[New sentences in revised manuscript]

“Notwithstanding, the renal markers of the RF mice remained significantly higher (p < 0.001 for plasma IS and BUN) and (p < 0.05 for CRE) compared to the control group. Notably, the PH-HPβCD-treated mice showed better recovery, with significantly lower plasma IS and BUN (p < 0.05) and a slightly lower plasma CRE compared to the RF mice. Representative kidney sections (Figure 4f) showed the expansion of the peritubular spaces and the beginning of cyst formation, indicative of tubulointerstitial fibrosis in the RF mice. The treated mice showed less severe cyst formation, with slightly lower total fibrosis area than the RF mice, although this was not statistically significant, as shown in Figure 4g.” (Page 8; Line 212-220).

[Original sentences]

“PH-HPβCD treatment reduced plasma IS and BUN and showed a strong tendency to reduce plasma CRE. However, there was no significant improvement in renal tissue fibrosis. In 5/6 nephrectomized mice treated with PH-HPβCD, there was only a marginal reduction in plasma IS and no significant improvement in renal failure markers or tissue fibrosis.”

[New sentences in revised manuscript]

“PH-HPβCD treatment reduced plasma IS and BUN and showed a tendency to reduce plasma CRE and renal tissue fibrosis. In 5/6 nephrectomized mice treated with PH-HPβCD, there was a tendency to reduce plasma IS, BUN, and plasma CRE. However, no significant improvements in these renal failure markers or tissue fibrosis were observed.” (Page 12; Line 335-339).

Comment 7: The evidence for causality in nephroprotection would be stronger if in vitro cell models were also included.

Response: We agree with the reviewer’s comment on using in vitro cell models. We have conducted a preliminary investigation into the effect of PB and PH on immortalized human proximal tubule (HK-2) cells exposed to adenine. Our findings have been inconclusive; PB seems to protect the cells, while PH has yielded conflicting results. However, as mentioned in the future direction of the study in the Discussion section, we intend to thoroughly investigate the effect of the compounds on the kidneys at a cellular level in our future research.

Comment 8: The absence of specific effects of HCBCD must be more thoroughly investigated. Additionally, there is a trend towards a decrease in BUN and creatinine levels. Is there any histological analysis available to support these findings?

Response: Thank you for pointing this out. The BUN and creatinine levels of the HPβCD-supplemented mice show a tendency to decrease despite a slight increase in plasma IS levels. Per the reviewer’s suggestion, we have included the histological analysis of the mice kidney sections in the supplementary file. The histological data reveals a marginal and statistically insignificant decrease in the fibrosis area of the HPβCD-supplemented mice compared to the RF mice. Thus, considering the biochemical and histological data, it is evident that the administration of HPβCD alone, at the specified dosage, does not result in a significant renoprotective effect.

The following modifications have been made in the revised manuscript to reflect these points:

[Original sentences]

“The experiment was repeated using HPβCD alone instead of the PH-HPβCD solid system to evaluate the effect of HPβCD. The results showed no significant differences in plasma IS, BUN, and plasma CRE compared to the RF group (Figure S2a-c).”

[New sentences in revised manuscript]

“The experiment was repeated using HPβCD alone instead of the PH-HPβCD solid system to evaluate the effect of HPβCD. The results (Figure S2a-e) indicate a tendency for the plasma IS to increase, whereas BUN and plasma creatinine tended to decrease. However, these were not significantly different from the RF mice. Furthermore, despite a slight decrease, there were no significant differences in the renal fibrosis area of the HPβCD-supplemented mice compared to the RF mice. These suggest that HPβCD alone, at the dose administered, does not result in a significant renoprotective effect.” (Page 6; Line 174-180).

Comment 9: A similar analysis using PH alone (e.g., via oral gavage) would be interesting to clearly delineate the specific nephroprotective effects of this metabolite.

Response: Thank you for the suggestion. Though our initial attempts to administer PH alone were unsuccessful due to its very viscous nature and poor solubility, we will repeat it according to the reviewer’s suggestion. 

Round 2

Reviewer 1 Report

Comments and Suggestions for Authors

The authors have fully responded to all of my questions 

Comments on the Quality of English Language

well prepare